# Panic Attack, Severe Hypophosphatemia and Rhabdomyolysis in the Setting of a Motor Functional Neurological Disorder

**DOI:** 10.3390/brainsci13050726

**Published:** 2023-04-26

**Authors:** Thibault Schneider, Nicolas Broc, Beatrice Leemann, Armin Schnider, Nicolas Nicastro

**Affiliations:** 1Division of Neurorehabilitation, Department of Clinical Neurosciences, Geneva University Hospitals, 1206 Geneva, Switzerland; 2Faculty of Medicine, University of Geneva, 1206 Geneva, Switzerland

**Keywords:** functional neurological motor disorder, panic attack, rhabdomyolysis

## Abstract

(1) Background: panic attack is often regarded as a benign disorder with variable physical and psychological symptoms. (2) Case Presentation: We here report the case of a 22-year-old patient known for an episode of motor functional neurological disorder a year earlier who presented a panic attack with hyperventilation causing severe hypophosphatemia and rhabdomyolysis, as well as mild tetraparesis. Electrolyte disturbances quickly resolved after phosphate substitution and rehydration. However, clinical signs suggesting a relapse of a motor functional neurological disorder appeared (improved walking with dual tasks). Diagnostic workup, including brain and spinal magnetic resonance imaging, as well as electroneuromyography and genetic testing for hypokalemic periodic paralysis, was unremarkable. Tetraparesis, lack of endurance, and fatigue eventually improved after several months. (3) Conclusions: the present case report highlights the intertwined relationship between a psychiatric disorder, leading to hyperventilation and acute metabolic disturbances, and functional neurological manifestations.

## 1. Introduction

Functional neurological disorder (FND) is defined as neurological symptoms incongruent and inconsistent with a somatic neurologic disease [1]. FND can present with a variety of neurological symptoms, including motor or sensory manifestations, as well as movement disorders or functional seizures. Other terminology has been used throughout history, e.g., hysteria and somatoform disorder [2]. Conversion disorder is currently used in DSM-V [3]. FND has an annual incidence of 12/100,000 and a prevalence of 50/100,000. Patients of any age can experience FND and a female predominance (60–75%) is observed [4,5]. While the actual prevalence of FND is difficult to assess due to a lack of high-quality studies, FND is considered among the main diagnoses retained in outpatient neurology clinics [6]. Three types of factors are usually associated with the onset and evolution of FND: risk factors (history of a psychological stressor, e.g., sexual abuse, psychiatric disorder), precipitating factors (e.g., an acute stress or structural injury) and perpetuating factors (such as misdiagnosis or lack of treatment) [7].

DSM-V clinical diagnostic criteria for FND require an altered voluntary motor or sensory function in addition to evidence of incompatibility between the symptoms and a recognized neurological or medical condition [8]. Symptoms are not better explained by another medical or mental disorder and are causing significant distress.

FND pathophysiology remains unclear; neuroimaging data suggest alterations of brain networks, especially limbic and sensorimotor circuits involved in multimodal integration and attention [9]. More specifically in motor FND, increased functional connectivity between the amygdala and motor control circuits has been observed, as well as decreased activity in striato–thalamo–cortical networks during voluntary movement [10].

## 2. Case Presentation

We describe the case of a 22-year-old athletic man without any remarkable medical history who presented two distinct episodes of FND. The first one manifested as tetraparesis, which occurred while the patient was enrolled in military service and subject to intense physical exercise and a stringent diet. Clinical presentation of this first episode included hyperventilation inducing hypocapnia then hypophosphatemia, though without rhabdomyolysis, as well as tetraparesis. Exhaustive investigations were performed, i.e., cerebral and spinal cord MRI results were unremarkable, while workup for myasthenia gravis was negative (normal nerve conduction studies and absence of acetylcholine receptor antibodies). Hypokalemic periodic paralysis was ruled out with short/long exercise testing and genetic analysis of SCN4A, KCNJ18, KCNJ2 and CACNA1S. In the presence of atypical neurological signs, motor FND was considered. In fact, the patient had a buckling gait (his knees were giving way) and his walk improved while performing a dual task. The patient underwent a 4-week in-patient rehabilitation program, followed by out-patient physical therapy and psychotherapy. We observed a complete recovery within 6 months.

The second episode of FND occurred a year later. The patient was addressed to the emergency department of our institution for a panic attack manifesting as acute hyperventilation and anxiety. He also reported paresthesia and tetany in his four limbs. The symptoms appeared as he was having a break while at work. It is unclear whether sensorimotor symptoms preceded or followed the panic attack. In the previous months, the patient reported being able to perform normal physical activity and had a balanced diet.

Physical examination revealed tetraparesis with muscle strength at 4/5 according to Medical Research Council criteria with Trousseau’s sign on the right hand, normal cranial nerves, no significant sensory deficit and normal deep tendon and flexor plantar reflexes. Assessment of gait and balance was not performed in the acute setting.

Arterial blood gas analysis showed acute respiratory alkalosis with pH = 7.57 (normal range (NR): 7.35–7.45), hypocapnia with PaCO_2_ = 19.2 mmHg (NR: 30–45 mmHg), PaO_2_ = 105 mmHg (NR: 82.5–105 mmHg), HCO_3_^−^ =17.7 mmol/L (NR: 22–26 mmol/L) and a −1.3 mmol/L base excess (NR: ±2–3 mmol/L). Blood chemistry revealed severe hypophosphatemia (0.2 mmol/L) and rhabdomyolysis with creatinine kinase 44,913 U/L, aspartate aminotransferase = 397 U/L and alanine aminotransferase = 133 U/L. Other blood analyses, including electrolytes (sodium, potassium, magnesium and calcium), vitamins (B1, B2, B6, B9 and B12), amino acids and TSH, were unremarkable. Urinalysis did not show any renal loss of phosphate. Nerve conduction studies and myography did not reveal any abnormality. The patient was hospitalized and treated with sodium phosphate and intravenous rehydration. Electrolytic disturbances resolved after 72 h. However, the patient still presented a slow walk and gait instability with positive clinical signs for motor FND. In fact, his walk improved with dual (motor or cognitive) tasks, as well as when the examiner maintained a rhythmic pattern through hand clapping. The patient was transferred to our neurorehabilitation division for intensive physical and occupational therapy. The patient also presented a lack of endurance with 6 min walking distance at 545 m, hand weakness as assessed by Jamar test (42 kg right and 36 kg left) and a lack of hand dexterity (nine-hole peg test evaluations: 27″ and 24″ for right and left hands, respectively). Concurrently, the patient had a weekly psychotherapeutic follow-up. After four weeks, the patient was dismissed and pursued out-patient physical therapy. He presented gradual improvement in his motor deficits at three months and was able to resume academic studies.

## 3. Discussion

The present case report highlights the severe acute consequences of hyperventilation due to panic attacks, leading to hypophosphatemia and rhabdomyolysis followed by a relapse of motor FND.

A panic attack is defined as a sudden onset of intense discomfort, anxiety or fear accompanied by somatic and/or emotional symptoms [3]. Panic attacks can be associated with panic disorder or agoraphobia and have a lifetime prevalence of 28.3% [11]. Treatment is based on psychotherapy, such as cognitive behavioral therapy, and pharmacological treatment (antidepressants or benzodiazepines). The management of acute episodes includes breathing control techniques and relaxation.

Hyperventilation is defined as breathing in excess of metabolic requirements, leading to a pulmonary gas imbalance with hypocapnia [12]. This in turn increases cellular oxygen consumption and pH according to the Henderson–Hasselbalch equation [12]. There is no acute compensatory mechanism, which leads to alkalosis with blood and cellular accumulation of H^+^ [13,14]. This changes the ionic charge of cellular enzymes, accelerating metabolic reactions and ATP demand. In addition, stimulation of glycolysis increases phosphate demand, with the latter being used as a metabolite and induced hypophosphatemia resulting in ATP deficiency [15]. Homeostasis between intracellular and extracellular compartments is impaired with the imbalance between Na^+^/K^+^ ATPase and Ca^2+^ ATPase pumps, leading to cell death and rhabdomyolysis [16].

The prevalence of hypophosphatemia and rhabdomyolysis in panic attacks is difficult to assess. Due to a quick resolution of symptoms, patients do not necessarily seek health facilities, and a medical screening with a chemistry panel is not always performed. However, several case reports have described metabolic disturbances associated with paresthesia and weakness following a panic episode [14,17].

The management of motor FND is based on an integrated multidisciplinary approach inspired by a biopsychosocial model. However, standardized and validated management protocols are currently lacking. Of note, there is no clear evidence of the benefit of pharmacological therapy, particularly regarding antidepressants (in the absence of depression) [18]. Psychotherapeutic support (e.g., using cognitive behavioral therapy or psychodynamic therapy) is recommended, possibly beginning in the early stage of the disease [1,19,20]. However, there is no evidence of superiority for a specific psychotherapeutic model [20]. Psychoeducation is paramount so that patients can understand what FND represents, as well as its management [21].

Rehabilitation programs for motor FND also include physical and occupational therapy [22,23]. As weakness due to both a defined neurological condition and FND share clinical similarities, standard motor rehabilitation techniques can be applied. One notable difference is that patients with FND have the intrinsic ability to move a weak limb; therefore, therapeutic strategies can implement distraction techniques so the patient does not focus on a weaker limb. Graded exercise is also likely to be helpful in allowing the patient to process a gradual improvement in motor capabilities [24,25]. Current recommendations are mostly based on expert consensus, and evidence-based standardized care is warranted [26,27]. The long-term outcomes of motor FND after in-patient rehabilitation have been assessed in several studies [24,28,29,30,31], with sustained functionality observed after several years of follow-up. Predictors of a better long-term clinical outcome include the absence of a comorbid psychiatric disorder [32], a shorter duration of symptoms prior to diagnosis, willingness to accept the potential reversibility of symptoms and employment before the onset of FND [25,33]. Of note, neither age nor sex were described as having a significant impact on prognosis [25].

Among promising therapeutic approaches, non-invasive brain stimulation (e.g., transcranial magnetic stimulation of the motor cortex) is currently being explored. Long-term benefits for FND have been observed with various interventional protocols [34,35,36].

Based on the case reported above, we can suggest some hypotheses regarding the factors potentially involved in the onset and relapse of FND. We were not able to identify specific risk factors, such as a known psychiatric condition or sexual abuse. In fact, no history of anxiety, depression or panic attack was described. The patient was very sporty and set himself high standards in terms of physical performance as he previously was a semi-professional snowboarder.

Regarding potential precipitating factors (such as acute stress or physical injury), we can highlight that FND relapse appeared when the patient experienced limb paresthesia, more or less simultaneously to the episode of panic attack. Therefore, it is difficult to disentangle whether FND preceded or followed psychiatric manifestations. Most probably, both the psychiatric and neurological manifestations enhanced each other. In addition, the patient was under a lot of stress as he was experiencing intense working activity (as a full-time waiter) and was contemplating resuming academic studies a few months later.

Among perpetuating factors, we can hypothesize that while a diagnosis of FND was clear for our rehabilitation team, other investigations were still being performed during the rehabilitation process, including genetic tests. This aspect might have put the patient in an uncomfortable situation, in which diagnosis and possibly clinical recovery were unclear.

In similar situations, it is crucial to perform a thorough clinical examination, including maneuvers for FND. In fact, the diagnosis of a clinically established FND relies primarily on recognizing the characteristic features (incongruencies, positive signs) of FND and not simply ruling out other neurological conditions. However, neurological conditions must be considered in the differential diagnosis [37]. First, a “bizarre” or atypical presentation can be observed in organic conditions, e.g., gait disturbances in the setting of dystonia, chorea or orthostatic tremor and fluctuating weakness in the case of myasthenia gravis. Second, FND can often co-exist with a defined neurological disorder, with up to 15–20% prevalence of FND in Parkinson’s disease, multiple sclerosis and epilepsy [38,39,40].

The co-occurrence of depression, anxiety and panic disorder is frequent in FND [41]. In addition, higher rates of childhood traumas (physical, emotional) were reported in patients with functional movement disorders compared to organic movement disorders and controls [42].

To the best of our knowledge, this is the first description to report the complex clinical interaction between psychiatric manifestations (panic attack with hyperventilation), major metabolic disturbances (hypophosphatemia and rhabdomyolysis) and motor FND. The initial presentation of hypophosphatemia, rhabdomyolysis associated with paresthesia and muscle weakness can be observed in the acute setting of a panic attack. However, the persistence of neurological symptoms after the correction of metabolic disturbances and clinical incongruities should lead the clinician to consider FND in the differential diagnosis.

## Data Availability

The present manuscript is based on clinical data of a single patient. Therefore, no formal analyses were performed.

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
