# Peer review of "Panic Attack, Severe Hypophosphatemia and Rhabdomyolysis in the Setting of a Motor Functional Neurological Disorder"

_brainsci, 2023, doi:10.3390/brainsci13050726_

Round 1

Reviewer 1 Report

The herein presented interesting case report shows the complex relationships between a psychiatric manifestation, its metabolic consequences and its functional neurological consequences.

What is missing in this case report is a hypothesis or at least a thorough discussion and literature review on how FNDs develop. Anxiety/panic disorder (does the patient have a panic disorder or only a single panic attack - please specify!) as a predisposing factor, then the experience of metabolic disturbances as the precipitating factor and one might suggest a malfunctioning processing and integration of these sensations in the brain. Any perpetuating factors?

The time line is confusing, please sum it up in one additional sentence.

Fortunately, the patient had a full recovery thanks to in- and out-patient physiotherapy. The guidelines recommand not only physiotherapy in FMDs but also psychoeducation, to allow the patient an understanding of FNDs and treatment of comorbidities. Please elaborate on this process or what has taken place, respectively. (Stone J, Carson A, Hallett M. Explanation as treatment for functional neurologic disorders. Handb Clin Neurol. 2016;139:543-553. doi: 10.1016/B978-0-12-801772-2.00044-8. PMID: 27719870.)

I recommand extensive editing of English language and style.

Author Response

Response to Reviewers for

Brainsci-2291132 - “Panic attack, severe hypophosphatemia and rhabdomyolysis in the setting of a motor functional neurological disorder”

Dear Dr. Gzielo,

We would like to thank the reviewers for their valuable comments and are pleased to submit our revised manuscript mentioned above for consideration in Brain Sciences. As proposed, the modifications in the manuscript are highlighted in revision mode. We also provided a clean version.

Reviewers' Comments to Authors

Reviewer #1

The herein presented interesting case report shows the complex relationships between a psychiatric manifestation, its metabolic consequences and its functional neurological consequences.

What is missing in this case report is a hypothesis or at least a thorough discussion and literature review on how FNDs develop. Anxiety/panic disorder (does the patient have a panic disorder or only a single panic attack - please specify!) as a predisposing factor, then the experience of metabolic disturbances as the precipitating factor and one might suggest a malfunctioning processing and integration of these sensations in the brain. Any perpetuating factors?

Authors’ response: thank you very much for your comment. The Introduction and the Discussion sections have been significantly extended in order to more thoroughly discuss the hypotheses on how FND develop, in the specific case of our patient and in general.

The time line is confusing, please sum it up in one additional sentence.

Authors’ response: the presentation of the case has been rearranged chronologically to make it clearer. Thank you for your valuable comment.

Fortunately, the patient had a full recovery thanks to in- and out-patient physiotherapy. The guidelines recommand not only physiotherapy in FMDs but also psychoeducation, to allow the patient an understanding of FNDs and treatment of comorbidities. Please elaborate on this process or what has taken place, respectively. (Stone J, Carson A, Hallett M. Explanation as treatment for functional neurologic disorders. Handb Clin Neurol. 2016;139:543-553. doi: 10.1016/B978-0-12-801772-2.00044-8. PMID: 27719870.)

Authors’ response: our patient had a weekly psychological follow-up and we have now added this important information in the case presentation. In addition, the publication from Stone et al. has been added to the references, among many others.

I recommand extensive editing of English language and style.

Authors’ response: thank you for your comment. Extensive editing has been provided to ensure a more comfortable reading.

Reviewer 2 Report

The authors reported a case of functional motor disorder in a patient with panic attack. I have some comments to the authors:

-       The authors should write a brief introduction on the topic, this is very important. The introduction should also cover general aspects of FND and FMD. For this purpose, the authors should include these relevant and important papers about FND and FMD:

Hallett M, et al. Functional neurological disorder: new subtypes and shared mechanisms. Lancet Neurol. 2022 Lidstone SC, et al. Functional movement disorder gender, age and phenotype study: a systematic review and individual patient meta-analysis of 4905 cases. J Neurol Neurosurg Psychiatry. 2022   

-       Can the authors provide a video recording of the patient?

-       The authors should better describe the positive signs in the case description paragraph.

-       The authors should mention the important overlap between FND and other neurological/psychiatric disorders.

-       Please try to expand the discussion exploring the possible relationship between the events that you found in this patient.

-       How are frequent hypophosphatemia and rhabdomyolysis during panic attack and tetraparesis? please specify

-    The authors should highlight the importance of considering FND during these clinical events.

Author Response

Response to Reviewers for

Brainsci-2291132 - “Panic attack, severe hypophosphatemia and rhabdomyolysis in the setting of a motor functional neurological disorder”

Dear Dr. Gzielo,

We would like to thank the reviewers for their valuable comments and are pleased to submit our revised manuscript mentioned above for consideration in Brain Sciences. As proposed, the modifications in the manuscript are highlighted in revision mode. We also provided a clean version.

Reviewers' Comments to Authors

Reviewer #2

The authors reported a case of functional motor disorder in a patient with panic attack. I have some comments to the authors:

-       The authors should write a brief introduction on the topic, this is very important. The introduction should also cover general aspects of FND and FMD. For this purpose, the authors should include these relevant and important papers about FND and FMD:

Hallett M, et al. Functional neurological disorder: new subtypes and shared mechanisms. Lancet Neurol. 2022 Lidstone SC, et al. Functional movement disorder gender, age and phenotype study: a systematic review and individual patient meta-analysis of 4905 cases. J Neurol Neurosurg Psychiatry. 2022   

Authors’ response: thank you for your comment, which was also raised by Reviewer #1. We have now provided a comprehensive introduction section discussing the definition and pathophysiology of FND. The paper from Hallett et al. has been added to the references.

-       Can the authors provide a video recording of the patient?

Authors’ response: I am sorry that we are not able to provide a video recording of the patient.

-       The authors should better describe the positive signs in the case description paragraph.

Authors’ response: a more precise description of the positive signs for FND has been added for both episodes.

-       The authors should mention the important overlap between FND and other neurological/psychiatric disorders.

Authors’ response: thank you for this valuable comment. We added a paragraph about differential diagnosis and overlap of functional neurological disorders and both neurological and psychiatric conditions.

-       Please try to expand the discussion exploring the possible relationship between the events that you found in this patient.

Authors’ response: the case presentation timeline has been reworked and expanded in order to provide more information about the onset, development and relapse of the patient.

-       How are frequent hypophosphatemia and rhabdomyolysis during panic attack and tetraparesis? please specify

Authors’ response: we were not able to assess the prevalence of metabolic disturbances and muscle weakness following a panic attack. Reasons for that have been added to the Discussion section. However, we included two additional publications which described neurological symptoms following a panic episode with hypophosphatemia.

-    The authors should highlight the importance of considering FND during these clinical events.

Authors’ response: thank you. We have stressed the importance of considering FND for similar situations.

Round 2

Reviewer 2 Report

The authors have addressed all the points.